# The Effect of Exposure to Noise during Military Service on the Subsequent Progression of Hearing Loss

**DOI:** 10.3390/ijerph18052436

**Published:** 2021-03-02

**Authors:** Brian C. J. Moore

**Affiliations:** Cambridge Hearing Group, Department of Psychology, University of Cambridge, Downing Street, Cambridge CB2 3EB, UK; bcjm@cam.ac.uk

**Keywords:** noise-induced hearing loss, military service, progression of hearing loss, effect of age

## Abstract

This paper reviews and re-analyses data from published studies on the effects of noise exposure on the progression of hearing loss once noise exposure has ceased, focusing particularly on noise exposure during military service. The data are consistent with the idea that such exposure accelerates the progression of hearing loss at frequencies where the hearing loss is absent or mild at the end of military service (hearing threshold levels (HTLs) up to approximately 50 dB HL), but has no effect on or slows the progression of hearing loss at frequencies where the hearing loss exceeds approximately 50 dB. Acceleration appears to occur over a wide frequency range, including 1 kHz. However, each of the studies reviewed has limitations. There is a need for further longitudinal studies of changes in HTLs over a wide range of frequencies and including individuals with a range of HTLs and ages at the end of military service. Longitudinal studies are also needed to establish whether the progression of hearing loss following the end of exposure to high-level sounds depends on the type of noise exposure (steady broadband factory noises versus impulsive sounds).

## 1. Introduction

Two common causes of hearing loss are exposure to high-level sound [1] and increasing age [2]. While there are standard methods for predicting how hearing loss will progress during prolonged exposure to high-level sound [3] and how hearing loss worsens with increasing age [4], the interaction of the two is poorly understood. Specifically, it is not clear whether and how exposure to high-level sound influences the progression of hearing loss after the exposure has ceased; does the exposure accelerate the subsequent progression of hearing loss, slow it, or have no effect? This issue is of practical consequence, especially for people who have been exposed to high-level sounds during military service. Some such people have near-normal hearing at the time of discharge from military service but develop hearing loss some time later. The question then arises: is the hearing loss partly caused by the prior noise exposure? This paper reviews the available evidence, and presents a re-analysis of the data reported by Macrae [5,6]. Evidence is presented that for frequencies where the noise-induced hearing loss (NIHL) at the end of military service is mild, exposure to noise during military service accelerates the subsequent progression of hearing loss. In contrast, for frequencies where the NIHL is moderate or severe at the end of military service, the prior noise exposure has no effect on or slows the subsequent progression of hearing loss.

## 2. Theoretical Background and Studies with Animals

Mild to moderate hearing loss as revealed in the audiogram is probably primarily a consequence of loss of function of the outer hair cells (OHCs) in the cochlea. Hearing loss up to approximately 55 dB may be caused exclusively by loss of OHC function [7,8]. Greater hearing losses probably involve dysfunction of other structures in the cochlea, including the inner hair cells (IHC). It is known from studies of animals that a certain amount of damage to the OHCs can occur with little or no change in the detection threshold [9,10,11]. Furthermore, substantial damage to the IHCs, or to the synapses that link the IHCs to the neurons that make up the auditory nerve, can occur with little effect on the detection threshold [12,13]. These findings can be described in terms of the concept of a “cochlear reserve”; the cochlea has a certain spare capacity and can sustain some damage without loss of function as revealed by the audiogram, but once the reserve is sufficiently depleted effects in the audiogram then become apparent.

If hearing loss approaching 55 dB is present at some frequencies at the end of noise exposure, this could be due to near-complete loss of function of OHCs. In this case, acceleration of the subsequent progression of hearing loss due to further OHC damage is not expected. However, damage to IHCs and/or synapses might contribute to the hearing loss, in which case there could be scope for accelerated progression. If the hearing loss at the end of noise exposure is slight or mild at some frequencies, then there is scope for acceleration of the subsequent progression of hearing loss at those frequencies due to further damage to OHCs, IHCs and/or synapses.

Direct evidence for acceleration of the progression of hearing loss following exposure to high-level noise has come from two studies using mice. Kujawa and Liberman [14] compared the progression of hearing loss with increasing age for non-exposed mice and mice exposed to an octave-wide band of noise (8–16 kHz) with a level of 100 dB SPL for two hours, at various different ages (4–124 weeks). The noise-exposed mice were held with unexposed cohorts for post-exposure times from 2 to 96 weeks. When evaluated 2 weeks after exposure, shifts in detection threshold up to 40–50 dB were found for animals that were young when exposed (4–8 weeks), but animals exposed at the age of 16 weeks or later showed almost no threshold shift at the same post-exposure time. However, when held for long post-exposure times, exposed animals showed greater hearing loss than age-matched non-exposed controls. The exposed animals showed substantial deterioration of cochlear neural responses and corresponding histological evidence of primary neural degeneration throughout the cochlea. These effects were particularly strong for young-exposed animals, but were observed for all noise-exposed animals held 96 weeks after exposure. The authors concluded that “Data suggest that pathologic but sublethal changes initiated by early noise exposure render the inner ears significantly more vulnerable to aging.”

In a second study using mice [15], cochlear aging was compared after two types of noise exposure: one producing permanent synaptic damage without hair cell loss and another producing neither synaptic damage nor hair cell loss. A control group with no exposure was also used. Loss of synapses between the IHCs and the primary auditory neurons is called synaptopathy and the subsequent neural degeneration is called neuropathy [16]. Adult mice were exposed to an octave-wide noise band (8–16 kHz) for two hours at either 100 or 91 dB SPL. The mice were evaluated from 1 h to approximately 20 months after exposure. Cochlear function was assessed via distortion product otoacoustic emissions [17] and auditory brainstem responses (ABRs). The 100 dB SPL noise led to synaptopathy and to threshold shifts of 35–50 dB 24 h after exposure. After 2 weeks, thresholds had recovered, but synaptic counts and ABR amplitudes at high frequencies were reduced by as much as 45%. With increasing age, synaptopathy in the noise-exposed mice worsened compared to controls and spread to lower characteristic frequencies. Neuropathy followed. Threshold shifts first appeared more than 1 year after exposure and by approximately 20 months thresholds were up to 18 dB greater for the group with synaptopathy than for the other two groups. OHC losses worsened over the same time frame. In contrast, the group that was exposed to the lower-level noise (91 dB SPL) showed temporary threshold shifts without acute synaptopathy and did not show acceleration of synaptic loss or cochlear dysfunction with increasing age up to approximately 1 year after exposure. The authors concluded that “Therefore, interactions between noise and aging may require an acute synaptopathy, but a single synaptopathic exposure can accelerate cochlear aging”.

Mice are known to be more sensitive to damage from noise exposure than humans. It has been estimated that noise levels would have to be 14 dB higher in humans to cause the same temporary threshold shift as in mice [18]. It seems reasonable to infer that a noise level of 114 dB SPL for two hours would be sufficient to lead to synaptopathy in humans, as it does in primates [19], and to cause an acceleration in the subsequent progression of hearing loss. Levels markedly exceeding this are often encountered during military service [20], especially when hearing protection is not used or is not properly fitted. Additionally, cumulative exposure durations would be much greater than two hours for people with several years of military service. Furthermore, military personnel often enroll when they are younger than 20 years of age, and the data from mice reviewed above suggest that the cochlea is more susceptible to noise damage for young than for older individuals [15]. Finally, it is known that the impulsive sounds that are often encountered in military service produce more hearing damage than steady noises of equal energy [21,22,23]. Overall, it seems clear that the sound exposures that typically occur during military service would in principle be sufficient to produce an acceleration of the subsequent progression of hearing loss in humans.

## 3. Studies with Humans

There are only a few studies that have directly examined whether and how noise exposure affects the progression of hearing loss in humans after the exposure has ceased. These studies are summarised in Table 1. Of these studies, four [24,25,26,27] were based on samples of the general population and presumably included only a small proportion of former military personnel. The other three [5,28,29] were based on former military personnel, mostly army personnel (with non-exposed comparison groups).

### 3.1. The Study of Gates et al. (2000)

Gates, Schmid, Kujawa, Nam and D’Agostino [24] examined changes in hearing threshold levels (HTLs) over a 15 year period for 203 men from the Framingham Heart Study cohort. The mean age at the first hearing test was 64 years (range 58–80 years). It was assumed that occupational and recreational noise exposure over the 15 years was minimal due to the age of the cohort. The authors argued that a local elevation (notch) of the audiometric thresholds for frequencies in the range 3–6 kHz was indicative of noise-induced hearing loss and used an objective method to detect such notches. A notch of 15–34 dB in the range 3–6 kHz was deemed a small notch (N1), while notches of 35 dB or greater were deemed large notches (N2). The absence of a notch for ears with elevations less than 15 dB was denoted N0. The presence and absence of notches was found to correspond to the individuals’ history of noise exposure. For group N2 compared to groups N0 and N1, there was less change over 15 years for frequencies where there was already hearing loss (3–6 kHz), but more change for the adjacent frequency of 2 kHz, where the initial hearing loss was small. The authors concluded that “These data suggest that the noise-damaged ear does not ‘age’ at the same rate as the non-noise damaged ear. The finding of increased loss at 2 kHz suggests that the effects of noise damage may continue long after the noise exposure has stopped. The mechanism for this finding is unknown but presumably results from prior noise-induced damage to the cochlea.”.

### 3.2. The Study of Rosenhall (2003)

Rosenhall [25] presented audiometric findings for people who had taken part in a population-based study of 70 year olds in Gothenburg, Sweden. The analysis was partly cross sectional and partly longitudinal. The results showed that the deterioration of the HTL between the ages of 70 and 75 years was greater for individuals with high prior noise exposure than for those with low prior noise exposure over a range of frequencies, and especially at 2 kHz. However, the rate of change in HTL at 4 kHz was not significantly different for the two groups. This is illustrated in Figure 1. The author concluded that “the age-related deterioration of the frequencies 1, 2, and 8 kHz is more pronounced in elderly men exposed to noise compared with those not exposed to noise”.

### 3.3. The Study of Lee et al. (2005)

Lee, Matthews, Dubno and Mills [26] studied longitudinal changes in HTLs for 188 older adults (91 females, 97 males). At the time of entry into the study, their ages ranged from 60 to 81 years, with a mean of 68 years. The duration over which each individual was studied varied from 3 to 11.5 years, with a mean of 6.4 years. The slope of a linear regression line was used to estimate the rate of change in HTLs at each frequency for each ear. A questionnaire was used to identify those individuals with a history of noise exposure. Overall, there was no large effect of noise history on the rate of change of HTLs with increasing age. However, the authors stated that most of the individuals in their study did not show clear notches in their audiograms, which would be indicative of NIHL [1,30,31]. A greater proportion of males than females reported noise exposure and males with a positive noise history showed significantly higher HTLs for frequencies from 2 to 8 kHz than males with a negative noise history. For the males, the rate of change of HTL with increasing age was greater for the noise-exposed group at 1 and 2 kHz, and greater for the non-exposed group at 6 and 8 kHz, although none of the differences was statistically significant. Overall, the results are broadly consistent with the idea that noise exposure accelerates the progression of hearing loss for frequencies where the hearing loss at the end of exposure is absent or mild, but slows the progression of hearing loss for frequencies where the hearing loss at the end of exposure is moderate or severe.

### 3.4. The Study of Hederstierna and Rosenhall (2016)

Hederstierna and Rosenhall [27] studied the age-related change in HTLs for individuals with and without self-reported previous occupational noise exposure, assessed using a single question. This was a prospective, population-based, longitudinal study of individuals who were assessed at the age of 70 years and were assessed again at the age of 75 years. The total number of participants was 1013, of whom 365 had been exposed to high-level noise for more than 10 years before the age of 70 years. Only the better-hearing ear was assessed for each participant, and participants were excluded if they had conductive hearing loss in the better-hearing ear. There were no significant differences in the rate of change of HTLs at any frequency between the noise-exposed and the non-exposed groups. However, it should be noted that the noise-exposed group did not display any notch or dip in the audiogram in the frequency range 3–6 kHz, which is often taken as indicating NIHL [1,30,31]. Additionally, the noise-exposed males had rather high HTLs for frequencies of 2 kHz and above at age 70 years; the mean HTLs were approximately 40, 65 and 70 dB HL at 2, 4, and 8 kHz, respectively. These HTLs are in the range where prior noise exposure would not be expected to produce substantial changes in the rate of progression of hearing loss with increasing age.

### 3.5. The Study of Macrae (1971)

Macrae [5] compared the HTLs of military veterans obtained close to the end of military service and after an interval of several years. To the author’s knowledge, this is the only longitudinal study that has focused on the effects of noise exposure during military service on the subsequent progression of hearing loss. Macrae [5] compared the rates of change of HTL at 1 and 4 kHz with increasing age with those expected for a non-exposed population, based on the data of Spoor [32]. Twenty years later [6] he compared the rates of change of HTL with those expected from the ISO standard that was current at the time [33]. Here, the rates of change of HTL are compared with those expected from ISO 7029 [4], which is a current standard based on a large population who were carefully screened to exclude noise-exposed individuals. The section headed “Scope” in [4] includes the statement: “The data are applicable for estimating the amount of hearing loss caused by a specific agent in a population. Such a comparison is valid if the population under study consists of persons who are otologically normal except for the effect of the specific agent. Noise exposure is an example of a specific agent”.

Macrae analysed the audiograms of 240 individuals who had hearing loss at 4 kHz at the end of military service. Selection criteria were: (1) no conductive component to the hearing loss; (2) at the end of military service, an HTL at 4 kHz 20 dB or more greater than predicted for their age, based on the data of Glorig and Nixon [34]; (3) as far as possible, an HTL at 1 kHz not more than 10 dB greater than predicted for their age, based on the data of Glorig and Nixon [34] (this restriction was relaxed for some of the older individuals, because very few aged over 65 years met the criterion); (4) No evidence of a hereditary or non-organic component of the hearing loss. In what follows, the HTLs measured at the end of military service are denoted initial measurements.

Macrae divided the individuals into age groups and tabulated the changes in HTL at 1 and 4 kHz from the initial measurement to the time of the second measurement for each group. The relevant time interval varied across groups and the groups themselves varied somewhat depending on the frequency of interest and age. To make it easier to compare across groups, the changes in HTL tabulated by Macrae have been converted to rates of change of HTL in dB/year. The results for the left ear at 1 kHz are shown in Table 2. The first three columns show, for each group, the mean age, the mean initial HTL, and the observed rate of change of HTL. The rate of change of HTL tended to increase with increasing age (and increasing hearing loss), but the progression was somewhat irregular. To smooth the data, a linear regression line was fitted to the rate of change as a function of age, and the fitted line was used to predict the rate of change for each group. The results are shown in column 4 of Table 2. Column 5 shows the rates of change expected for non-noise exposed males of the same mean age based on ISO 7029 [4]. It is clear that the fitted observed rate of change exceeds the rate expected from ISO 7029 [4] for every age group. In other words, the progression of hearing loss with increasing age was greater for the noise-exposed group than expected for a non-noise exposed population.

Table 3 shows the results of a similar analysis for the right ear at 1 kHz. The pattern of the results is similar to that for the left ear. For every age group, the fitted observed rate of change of HTL exceeds the rate expected from ISO 7029 [4].

Table 4 shows the results of a similar analysis for the left ear at 4 kHz. Note that the HTLs at the end of military service were much higher than the HTLs at 1 kHz, which partly reflects the selection criteria. Even the youngest age group had a mean HTL of 52 dB HL. The pattern of the results differs from that for 1 kHz. For the groups with a mean age below 60 years, and a mean initial HTL below 60 dB HL, the fitted observed rate of change exceeds that expected from ISO 7029 [4]. For the groups with a mean age above 60 years, and a mean initial HTL above 60 dB HL, the fitted observed rate of change is below that expected from ISO 7029 [4].

Table 5 shows the results of a similar analysis for the right ear at 4 kHz. The results are similar to those in Table 3, except that for the groups over 60 years of age, with initial HTLs above 60 dB HL, the observed rates of change based on the linear fit are similar to the rates of change predicted from ISO 7029 [4].

The results in Table 2, Table 3, Table 4 and Table 5 are summarised in Figure 2. The fitted regression lines (thick grey lines for the observed rate of change and thin black lines for the rate predicted from ISO7029) give a clear visual impression of the differences between the observed and expected rates.

Macrae [5] also grouped individuals by the initial HTL at 4 kHz (which ranged from 40 to 80 dB HL). A similar analysis to that presented above showed that, for both ears, the line fitted to the observed rate of change of HTL had a slope close to but slightly below 1, i.e., the rate of change of HTL decreased slightly with increasing HTL at the end of military service. In contrast, based on the age range of each HTL group, the rate of change predicted from ISO 7029 increased with increasing HTL at the end of military service. For initial HTLs of 50 dB or less, the (fitted) observed rate of change was greater than predicted from ISO 7029, while for initial HTLs of 55 dB or more the (fitted) observed rate of change was similar to or less than predicted from ISO 7029.

Overall, the results of Macrae are consistent with the idea that for frequencies for which the hearing loss at the end of noise exposure is up to 50 dB, the progression of hearing loss following the end of the exposure is greater than would be expected from age alone. However, for frequencies for which the hearing loss at the end of noise exposure is 55 dB or more, the progression is similar to or less than expected from age alone. The accelerated progression produced by noise exposure, when it occurs, appears to be related more to the HTL at the end of the noise exposure than to age. This conclusion is supported by the observation that at 1 kHz acceleration occurred over the whole age range studied. It is noteworthy that acceleration occurred at 1 kHz, since it is often assumed that noise exposure has little effect at this frequency [30,35]. However, other more recent data support the idea that noise exposure during military service can have effects over a wide frequency range, including 1 and 8 kHz [36,37,38].

### 3.6. The Study of Xiong et al. (2014)

Xiong, Yang, Lai and Wang [28] performed a cross-sectional study using two groups, each containing 109 men. The two groups were matched in age, with a mean age of approximately 57 years (age range 55 to 60 years). Group I comprised military veterans who had all been exposed to the sound of shooting from rifles without hearing protection during a war. At the time of discharge, they were aged 23–25 years. All had normal hearing (HTLs ≤ 20 dB HL at all audiometric frequencies) at the end of military service. Group II comprised men with no military experience randomly chosen from a health examination center. Group II had no history of exposure to high-level sounds. Inclusion criteria for both groups were: (1) no history of ear disease; (2) no history of ototoxic drugs; (3) no familial history of ear diseases; (4) no diabetes, hyperlipidemia, cerebrovascular disease or other systemic disease. In addition, Group I met the following criteria: (1) no hearing loss, no tinnitus and other inner ear-related symptoms at the end of military service; (2) no occupational or non-occupational noise exposure after the end of military service.

Mean HTLs for the two groups, and differences across groups, are shown in Table 6. For frequencies up to 2 kHz, the noise-exposed group actually had slightly better HTLs than the non-exposed group, but the differences were not statistically significant. The small differences might indicate that the groups were not well matched in terms of factors other than age, such as general cardio-vascular fitness or history of smoking or alcohol consumption [39]. Alternatively, or in addition, the military veterans probably had their audiograms assessed regularly during military service, so they may have been more practiced at detecting soft sounds, leading to better HTLs. In any case, the noise-exposed group had clearly and significantly (*p* < 0.01) higher HTLs at 4, 6, and 8 kHz than the non-exposed group, consistent with the idea that the noise exposure during military service accelerated the subsequent progression of hearing loss.

Some limitations of this study should be noted. Firstly, the authors did not present the audiograms obtained for Group I at the end of military service, and the audiograms for Group II were not measured when they were 23–25 years old, so it is not clear how well the two groups were matched in HTL at 4, 6, and 8 kHz at the time when Group I were discharged from military service. Secondly, because Group I were selected to have HTLs within the “normal” range at the end of military service, while many of the military personnel who fought in the same war had hearing loss at the end of military service [28], Group I may represent those with “tough” ears, that are more resistant than average to the effects of noise, or may represent those who had lower than average noise exposure during military service.

### 3.7. The Study of Kim et al. (2017)

Kim, Lim, Kim and Park [29] studied the long-term effect of exposure to noise during military service by comparing the age-related change in HTL for groups with and without prior exposure to noise during military service, using a cross-sectional design. Individuals were excluded who had known causes of hearing loss other than noise exposure or had asymmetric audiograms. Group I comprised 3163 individuals with noise exposure during military service and Group II comprised 916 individuals without such exposure. The two groups had a wide range of ages, but most had ages in the range 30 to 69 years. The annual threshold deterioration rates were faster for the noise-exposed group than for the control group at 1, 2 and 4 kHz, as shown in Figure 3. Again, these results suggest that noise exposure during military service can accelerate the subsequent progression of hearing loss at some frequencies.

## 4. Discussion

Overall, the results of the three studies that focused on the effects of noise exposure during military service support the idea that such noise exposure can accelerate the subsequent progression of hearing loss at frequencies where the hearing loss at the end of military service is up to approximately 50 dB. For greater hearing losses, the prior exposure seems to have little effect or even slows the progression of hearing loss, consistent with the idea, described in Section 2, that when the HTL at a given frequency exceeds 55 dB HL, the OHCs are likely to be largely dysfunctional, so there is little scope for acceleration of hearing loss by further damage to the OHCs; any further hearing loss probably depends on dysfunction of IHCs, synapses, or other structures.

Each of the studies on the effects of noise exposure during military service had limitations. Two of the studies [28,29] had cross-sectional designs, which are more susceptible to bias effects than longitudinal designs. The single study that had a longitudinal design [5] assessed longitudinal changes only at 1 and 4 kHz and also selected individuals who had substantial hearing loss at 4 kHz at the end of military service. Clearly, further longitudinal studies are needed of changes in HTLs over a wide range of frequencies and including individuals with a range of HTLs and ages at the end of military service.

It is of interest that in the study of Macrae [5], there was no clear difference in HTLs between the left and right ears of the noise-exposed individuals. This contrasts with several more recent studies showing that exposure to noise during military service results, on average, in greater hearing loss for the left ear than for the right ear, although some individuals show the opposite pattern [37,38,40,41]. The difference across studies may be partly a consequence of an increase over time in the sound levels produced by military weapons and by the characteristics of those weapons. Specifically, Lowe and Moore [38] suggested that greater hearing loss for the left ear than for the right ear is often associated with use of the SA80 rifle. This, by design, can only be fired from the right shoulder. As a result, the right ear is tilted away from the muzzle and is partially shielded by the head from the sound emitted by the muzzle [42]. Another possible reason for the lack of ear asymmetry in the study of Macrae [5] is that, as far as possible, the selected individuals met his criteria for both ears, and this would have excluded individuals with near-normal hearing in one ear. Kim, Lim, Kim and Park [29] specifically excluded individuals with asymmetric hearing loss from their study. It would be of interest in future studies to assess the progression of hearing loss for individuals who have asymmetric loss at the end of military service and to compare the progression for the worse ear and for the better ear.

It is unclear whether the progression of hearing loss following the end of exposure to high-level sounds differs depending on the type of noise exposure. One might expect this to be the case, since the characteristics of NIHL produced by steady noises of the type encountered in factories are different from the the characteristics of NIHL produced by the impulsive types of sounds encountered in military service, which is denoted hereafter M-NIHL. Specifically: (1) steady broadband noises lead to a characteristic dip or notch in the audiogram, typically centered at 4 kHz and with little hearing loss at 1 kHz and below or at 8 kHz [1,31], whereas M-NIHL is associated with a wide range of patterns of hearing loss, sometimes with maximum loss at 8 kHz and often with some hearing loss at 1 or even 0.5 kHz [36,37,38]; (2) steady broadband noises usually lead to hearing loss that is symmetric across the ears, consistent with the symmetric exposure, whereas, as mentioned earlier, M-NIHL is often asymmetric across the two ears [37,38,40,41].

The population studies reviewed in Section 3.1, Section 3.2, Section 3.3 and Section 3.4 probably included only a small proportion of people who had served in the military; the noise exposure of those populations was probably mainly from working in factories and leisure activities. In any case, those studies focused mainly on older people who already had substantial hearing loss at high frequencies. Thus, the data from those studies are not suitable for assessing whether exposure to high-level noise affects the progression of hearing loss after the exposure has ceased for individuals who are young and/or have little or only mild hearing loss at the end of noise exposure.

It should be emphasized that all of the studies reviewed here focused on average or median changes in HTL over time. In all studies, substantial individual variability was noted. It is likely to be the case that some individuals with M-NIHL will show little or no acceleration of the progression of hearing loss following the end of military service, while others will show very marked acceleration. Such individual differences might reflect differences in the underlying physiological damage to the auditory system, genetic makeup, cardiovascular health, lifestyle, and diet [43,44].

## 5. Conclusions

The evidence reviewed here is consistent with the idea that exposure to the high-level sounds that are encountered during military service accelerates the progression of hearing loss after the exposure has ceased for frequencies where the hearing loss is absent or mild (HTLs up to approximately 50 dB HL), but has no effect on or slows the progression of hearing loss for frequencies where the hearing loss exceeds 50 dB. However, each of the studies reviewed has its limitations. Further longitudinal studies are needed of changes in HTLs over a wide range of frequencies and including individuals with a range of HTLs and ages at the end of military service. Further longitudinal studies are also needed to establish whether the progression of hearing loss following the end of exposure to high-level sounds differs depending on the type of noise exposure.

## Figures and Tables

**Figure 1 ijerph-18-02436-f001:**
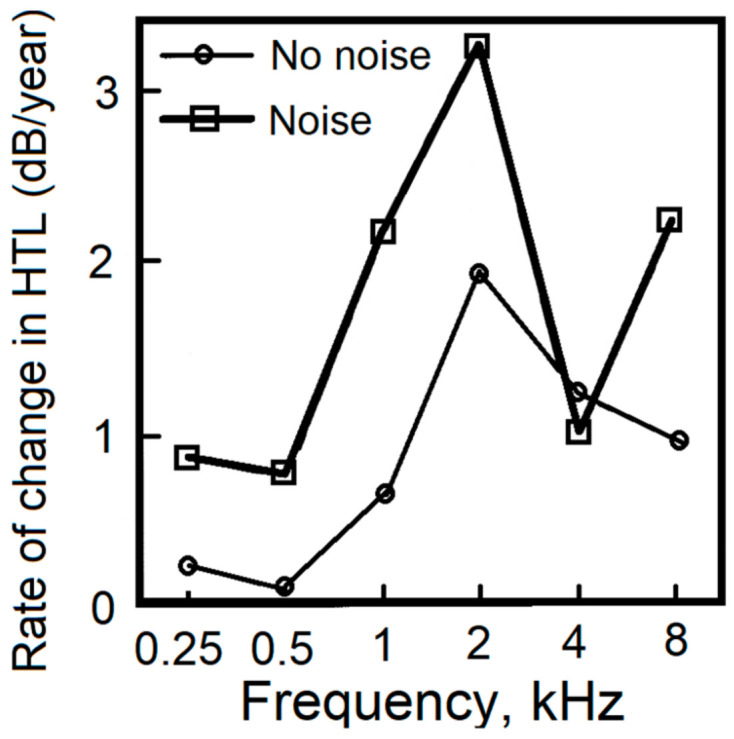
Rate of change of HTLs (dB/year) for groups with high prior noise exposure (thick line with squares) and low prior noise exposure (thin line with circles). Redrawn from Rosenhall [25].

**Figure 2 ijerph-18-02436-f002:**
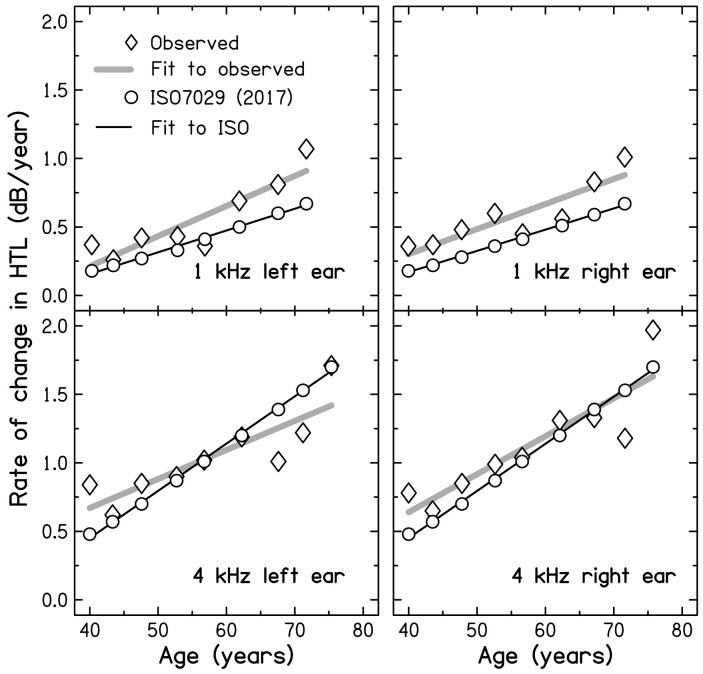
Rate of change of HTLs (dB/year) plotted against age group, as extracted from the data of Macrae [5] (triangles), based on linear fits to the data of Macrae (thick gray lines), based on ISO 7029 [4] (circles), and based on linear fits to ISO 7029 (thin black lines). Each panel shows results for one ear and one frequency, as indicated in the keys.

**Figure 3 ijerph-18-02436-f003:**
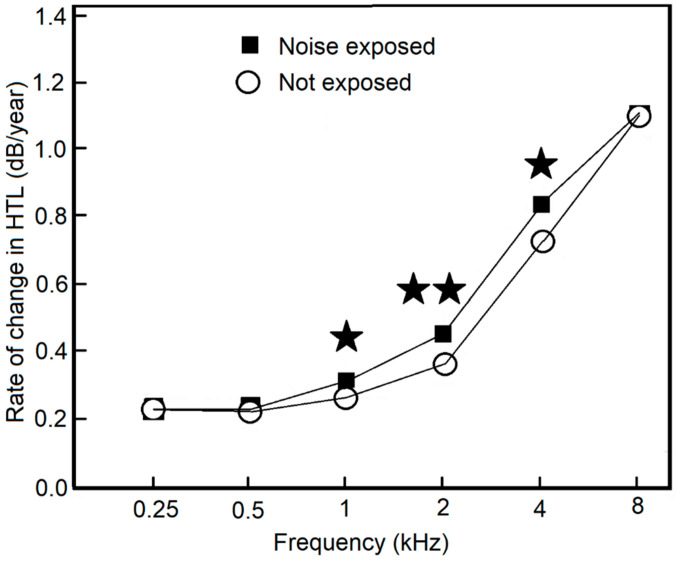
Rate of change of HTLs (dB/year) as a function of test frequency for Group I (exposed to noise during military service) and Group II (non-exposed control). ★ and ★★ indicate *p* < 0.05 and *p* < 0.01, respectively. Redrawn from [29].

**Table 1 ijerph-18-02436-t001:** Studies of the effect of noise exposure on the subsequent progression of hearing loss in humans.

Authors	Date	Design	Population	Age at Start
Gates et al.	2000	Longitudinal	General	58–80 years
Rosenhall	2003	Longitudinal and cross sectional	General	Mean of 70 years
Lee et al.	2005	Longitudinal	General	60–81 years
Hederstierna and Rosenhall	2016	Longitudinal	General	Mean of 70 years
Macrae	1971	Longitudinal	Military veterans	30–74 years
Xiong et al.	2014	Cross sectional	Military veterans and non-exposed controls	55–60 years
Kim et al.	2017	Cross sectional	Military veterans and non-exposed controls	30–69 years

**Table 2 ijerph-18-02436-t002:** Data for the left ear at 1 kHz for each age group of Macrae [5]. The columns show, from left to right: the mean age of the group, the mean initial HTL at 1 kHz for the left ear for each group, the observed rate of change of HTL for that group, the observed rate of change based on a linear fit to the data across groups; and the rate of change expected for non-noise exposed males of the same mean age based on ISO 7029 [4].

Mean Age, Years	HTL at End of Military Service	Rate of Change of HTL, dB/Year		
		Observed	Observed Based on Linear Fit	Based on ISO7029 (2017)
40.3	4.0	0.37	0.22	0.18
43.45	5.0	0.26	0.29	0.22
47.6	2.8	0.42	0.38	0.27
52.85	5.4	0.43	0.50	0.33
56.85	4.5	0.36	0.59	0.41
61.9	7.5	0.69	0.70	0.50
67.55	10.5	0.81	0.82	0.60
71.7	14.5	1.07	0.91	0.67

**Table 3 ijerph-18-02436-t003:** As Table 2, but for the right ear at 1 kHz.

Mean Age, Years	HTL at End of Military Service	Rate of Change of HTL, dB/Year		
		Observed	Observed Based on Linear Fit	Based on ISO7029 (2017)
39.95	5.0	0.36	0.30	0.18
43.55	4.3	0.37	0.37	0.22
47.75	3.3	0.48	0.45	0.28
52.6	3.5	0.60	0.54	0.36
56.55	3.2	0.45	0.61	0.41
62.45	8.9	0.56	0.72	0.51
67.15	7.3	0.83	0.80	0.59
71.6	14.5	1.02	0.88	0.67

**Table 4 ijerph-18-02436-t004:** As Table 2, but for the left ear at 4 kHz.

Mean Age, Years	HTL at End of Military Service	Rate of Change of HTL, dB/Year		
		Observed	Observed Based on Linear Fit	Based on ISO7029 (2017)
40	52.0	0.84	0.67	0.48
43.35	54.8	0.62	0.74	0.57
47.6	57.2	0.85	0.83	0.70
52.7	49.6	0.90	0.94	0.87
56.75	55.7	1.02	1.03	1.01
62.25	65.5	1.19	1.14	1.20
67.6	62.5	1.01	1.26	1.39
71.2	67.5	1.22	1.33	1.53
75.35	77.5	1.71	1.42	1.70

**Table 5 ijerph-18-02436-t005:** As Table 2, but for the right ear at 4 kHz.

Mean Age, Years	HTL at End of Military Service	Rate of Change of HTL, dB/Year		
		Observed	Observed Based on Linear Fit	Based on ISO7029 (2017)
40.0	55.5	0.78	0.64	0.48
43.5	55.0	0.65	0.73	0.57
47.8	58.1	0.85	0.85	0.70
52.65	54.3	0.99	0.99	0.87
56.6	53.8	1.04	1.10	1.01
62.1	63.2	1.31	1.25	1.20
67.16	63.0	1.33	1.39	1.39
71.65	69.2	1.18	1.51	1.53
75.75	79.4	1.97	1.63	1.70

**Table 6 ijerph-18-02436-t006:** Mean HTLs obtained by Xiong, Yang, Lai and Wang [28] for a military noise-exposed group and a non-exposed group. The bottom row shows the difference in HTLs between the two groups at each frequency.

	Frequency, kHz					
	0.5	1	2	4	6	8
Military exposed, dB HL	20.3	22.3	19.5	28.4	37.7	45.9
Non-exposed, dB HL	21.6	23.6	22.4	23.7	22.5	25.1
Difference, dB	−1.3	−1.3	−2.9	4.4	15.2	20.8

## Data Availability

All data presented in this paper are available in the published studies referred to in the text.

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
