# Peer review of "The Effect of Exposure to Noise during Military Service on the Subsequent Progression of Hearing Loss"

_ijerph, 2021, doi:10.3390/ijerph18052436_

Round 1

Reviewer 1 Report

This read like an editorial paper. No methodology, or no explanation how the included studies were selected, inclusion and exclusion criteria. I'm aware that this is a reanalysis of available data, but there must still be a criterion against which the study were selected also, providing a table with all the selected studies would have been helpful.

Author Response

All relevant studies were included. A table with the studies has been added.

Reviewer 2 Report

The author has done a superb and critical review of noise induced hearing loss in the military and its impact on progression of hearing loss with aging.

Author Response

No response needed.

Reviewer 3 Report

A minor one: You use the expression "intense noise" throughout the paper: there is not such a thing. There is high level noise.Please replace.

More important:

a) 3.2 (Fig 1). Please elaborate on the values at 4 KHz. Looks very strange. Some mistake may be?

b) 3.4 Please expand. Your review is too sketchy without getting in that many details as in 3.5

c) There are two 3.5 sections. Please correct.

d) Excellent Conclusion Section!

Author Response

I think that "intense noise" is a well-recognised term, but it has been replaced with "high level noise".

The values at 4 kHz have been correctly plotted. It is now noted in the text that the values at 4 kHz did not differ significantly for the grups with high and low noise exposure.

The description of the study in section 3.4 has been expanded.

The section numbering has been corrected.